# Modelling approaches for predicting the distribution of skin NTDs: A systematic review

**Mesoud A. Bushara**[ID][1]*, **Rowa Hassan**[2], **Rana Ahmed**[1,3], **Jorge Cano**[4], **Gail Davey**[2], **Eltayeb Ganawa**[1], **Hope Simpson**[2,5]

**1** University of Khartoum, Faculty of Geographical and Environmental Sciences, Khartoum, Sudan, **2** Centre for Equitable Global Health Research, Brighton and Sussex Medical School, Brighton, United Kingdom, **3** Converge: Centre for Chronic Disease and Population Health Research, School of Population Health, RCSI University of Medicine and Health Sciences, Dublin, Ireland, **4** World Health Organisation, Brazzaville, Republic of Congo, **5** London School of Hygiene and Tropical Medicine, London, United Kingdom

* mesoudozail@gmail.com

## Abstract

### Background

Skin neglected tropical diseases (NTDs) such as cutaneous leishmaniasis, lymphatic filariasis, mycetoma, and podoconiosis affect millions in endemic regions, but are under-recorded despite causing significant burdens. Predictive modelling has been used to estimate the distribution and prevalence of some of these diseases, and predictions may be useful for identifying at-risk populations and guiding interventions. This review synthesises the literature on modelling approaches to predict skin NTD distributions, aiming to identify prevalent methodologies, evaluate their strengths and limitations, highlight research gaps, and provide recommendations for enhancing their utility.

### Methods

We conducted a systematic literature review from three databases and included studies published from 2000-2024. Studies were included if they employed statistical models or machine learning algorithms to predict the distribution of skin NTDs. Two independent reviewers screened titles, abstracts, and full texts. Data extracted included disease, study region, source of epidemiological data, model types and predictors.

### Results

From 2,870 retrieved records, 68 met the inclusion criteria. The most modelled skin NTDs were cutaneous leishmaniasis (n = 26) and lymphatic filariasis (n = 18). Geostatistical modelling was the most common approach, followed by ecological niche modelling, with MaxEnt and generalised linear models constituting the predominant model

**Data availability statement:** All data underlying the findings are fully available within the manuscript and its Supporting Information files. Extracted data from included studies and analysis scripts are provided as supplementary materials.

**Funding:** This research is funded by the National Institute for Health and Care Research (NIHR) (16/13629 to GD) Global Health Research Unit Programme. The views expressed are those of the author(s) and not necessarily those of the NIHR or the Department of Health and Social Care. The funders had no role in study design, data collection and analysis, decision to publish, or preparation of the manuscript.

**Competing interests:** The authors have declared that no competing interests exist.

types. Common environmental covariates included climate, land cover and land use, elevation, and soil data. The types of epidemiological data varied, with many studies relying on passive surveillance and pseudoabsence data. The risk of bias was high among ecological niche models.

## Conclusions

Environmental and geostatistical models can inform targeted interventions for skin NTDs, aiding efficient resource allocation and public health planning. However, data limitations, especially the absence of true absence data, underreporting and variations in surveillance sensitivity, can reduce model accuracy and undermine decision-makers' confidence. Future studies should focus on incorporating information about case identification into modelling frameworks, including a broader spectrum of environmental and socio-economic determinants, and ensuring validation across diverse geographic regions.

### Author summary

Skin diseases caused by neglected tropical diseases (NTDs) such as cutaneous leishmaniasis, lymphatic filariasis, mycetoma, and podoconiosis affect millions of people, mostly in poorer regions. These conditions can cause disability, stigma, and poor mental health, yet they are often underreported and not well mapped. Predictive modelling, which combines health and environmental data, can help show where these diseases are likely to occur and guide public health responses. In this study, we reviewed published research that used modelling to predict the distribution of skin NTDs. We found that most studies focused on a few diseases and used statistical or environmental models. While useful, these models often struggled with limited or incomplete data. Our review highlights the need for better surveillance and improved data to make models more accurate. Stronger models can help health programs direct resources more efficiently and support efforts to control and eventually eliminate these diseases.

## Introduction

Skin neglected tropical diseases (NTDs) are a group of conditions that fundamentally affect the skin, cause significant morbidity, and can lead to disabilities, stigmatization, and negative mental health outcomes [1]. The skin NTDs recognized by the World Health Organization (WHO) are Buruli ulcer (BU), cutaneous leishmaniasis (CL), Hansens disease (leprosy), lymphatic filariasis (LF), mycetoma, chromoblastomycosis, and other deep mycoses, onchocerciasis (river blindness), post-kala-azar dermal leishmaniasis, scabies and other ectoparasitoses and yaws [2, 3]. Though the WHO does not include podoconiosis in this list, it recommends its integration with other skin

NTDs for control and management. Dracunculiasis is another NTD affecting the skin but not included within the WHO strategic framework for skin NTD control [4].

## Global frameworks for skin NTD control

The WHO has set targets for controlling and eliminating NTDs in a road map published in 2021, highlighting the importance of robust surveillance systems to track the prevalence and distribution of skin NTDs, along with strategies for mapping disease hotspots and planning interventions [3]. The WHO Skin NTD Framework promotes integrated management strategies, emphasizing surveillance, diagnosis, treatment, community engagement, and monitoring [ 2,5]. Effective targeting and integration of prevention, diagnosis, and treatment requires a thorough understanding of the geographical distributions of skin NTDs [6, 7].

## Environmental determinants of Skin NTDs

NTDs exhibit significant variation in prevalence across different regions, influenced by factors such as climate, ecology, and socioeconomics [4, 8]. Those which are vector borne (CL, LF, onchocerciasis) have strong environmental determinants, limiting the distributions of the insects that carry them [9], as does podoconiosis, which only occurs in certain environments characterized by particular soil types, altitude, and precipitation conditions [1]. Others such as BU and mycetoma are transmitted from the environment and are known to be influenced by certain environmental factors, though evidence for this is more limited [10]. Leprosy, scabies and yaws are directly transmitted from person-to-person, and there is limited evidence for an effect of the environment on transmission, although scabies has been reported to be seasonal in some settings, while in others no meaningful seasonal influence has been observed [11, 12].

## Existing data on Skin NTDs

For diseases like BU, leprosy, and scabies, data are often collected through passive surveillance. However, this approach has limitations resulting from poor healthcare access, low diagnostic capacity, and underreporting in endemic or high-prevalence areas, leading to underestimation of disease burdens [8, 13]. Moreover, there is a significant discrepancy in NTD cases recorded by programmes and reported to WHO [14]. Additionally, some affected countries lack the necessary surveillance systems to effectively track and document all cases, further contributing to the imprecision in global estimates of disease burdens. Active case detection is used for some skin NTDs, such as LF, through the Global Programme to Eliminate Lymphatic Filariasis (GPELF), with limited searches for BU and others [15]. Surveys for onchocerciasis (nodule prevalence) and podoconiosis (mainly for research) highlight the broader issue: high-quality survey data is scarce due to the high costs involved [16]. Consequently, current data often does not accurately represent the true distribution of skin NTDs, being influenced by varying levels of surveillance intensity across regions.

## Predictive modelling approaches

In view of the gaps and biases affecting current data on skin NTDs, the last two decades have seen development of various models to predict the geographical distribution of certain skin NTDs at fine scales, providing estimates of prevalence or suitability for target diseases in regions with limited data [17]. The approaches used can be categorized broadly as ecological models, which link occurrence to environmental conditions, and geostatistical models, which model spatial dependency to predict prevalence between sampled locations, and can also incorporate environmental relationships [18].

Ecological models predict where organisms are likely to occur based on field observations and environmental factors. A variety of terms are used to describe overlapping approaches to this aim, including ecological niche modelling (ENM), species distribution modelling (SDM), and risk mapping. While the terms ENM and SDM are sometimes used interchangeably, there is growing recognition that they cover distinct, though related concepts [19]. According to this distinction, ENM

predicts potential distributions based on fundamental niches, encompassing areas where a species/disease is found (the realised niche), and those with suitable habitat in which the target is non-existent (the unfilled niche), without distinguishing between these scenarios. In contrast, SDM is understood to predict actual distributions in geographic, rather than environmental space, based on both suitability and dispersion and colonisation potential [20, 21] or transmission models to generate risk maps which also account for the likelihood of contact and susceptibility [22]. However, risk mapping is also used to describe a variety of other approaches, which can include spatial effects. For the purposes of this review, we use the term "ecological models" to cover those which use statistical and/or machine learning approaches to relate occurrence of a target (meaning a disease or species) to environmental conditions, without incorporating spatial effects.

Since obtaining reliable data on disease or species non-occurrence is challenging, researchers often employ pseudo-absences or background points, representing locations where occurrence is unknown, but which are treated as model negatives. These enable comparison of environmental conditions at presence and background locations, allowing statistical relationships between occurrence and predictors to be fit and projected [23]. Ecological models do not typically quantify or account for spatial autocorrelation in occurrence records. Spatial autocorrelation, or spatial dependency, refers to the degree to which values of a particular measure (for instance suitability) are correlated with those at nearby locations. This effect can invalidate the assumptions of standard regression approaches and result in inaccuracies [24].

In contrast to ecological models, geostatistical models explicitly integrate and account for spatial dependency between nearby points, resulting in more precise and spatially consistent predictions [25]. These models have proven pivotal in charting the distribution of NTDs controlled by preventive chemotherapy (PC-NTDs) [26, 27], and have recently been applied to guide interventions against them [28, 29]. The application of geostatistical models to PC-NTDs reflects the availability of systematically collected survey data for these diseases, allowing prevalence to be modelled as a spatially continuous outcome. For diseases controlled mostly through individual based management, surveillance data tends to comprise reports of routinely identified cases. As a result, occurrence is represented by discrete points, and absence locations are not well defined [30].

The objective of this literature review is to describe, compare, and evaluate existing modelling approaches used to predict the distribution of skin NTDs. We aim to synthesize relevant studies to identify common modelling approaches, evaluate their strengths and limitations, and propose recommendations for improving skin-NTD prediction accuracy and utility.

## Methodology

### Search strategy

Our systematic search encompassed a combination of three databases: PubMed, Ovid, and Web of Science. The search strategy included a combination of terms and free-text keywords related to skin NTDs, prediction, and modelling, used the search string included in (Table 1). We restricted the search to studies published from 2000 onwards, as this period marks the expansion of spatial epidemiology and the widespread availability of geospatial tools, digital disease data, and environmental datasets. Earlier studies rarely employed modern predictive modelling approaches or accessible geospatial platforms, making them less comparable to contemporary methods. This restriction ensured methodological consistency and relevance to current public health applications.

The search was most recently updated in September 2024. Studies in English, French, Portuguese, Spanish, and Turkish were included, chosen for their relevance to endemic regions and the expertise of the review team. Non-English/French articles were translated using Google Translate. The auto-resolve feature in Rayyan [31] was used to remove duplicates. We also reviewed the references of key publications to identify relevant articles that may have been missed from the database search.

### Eligibility criteria

Studies were included or excluded based on predefined criteria (Table 2). Eligible studies were those that used modelling approaches to predict the distribution of skin NTDs in humans at a fine-scale (pixel- or lowest administrative

**Table 1. Example search string used for PubMed.**

| Component | Search Terms | Combine with |
|---|---|---|
| Diseases of interest | ("Buruli ulcer"[Title/Abstract] OR "Bairnsdale ulcer"[Title/Abstract] OR "ulcerans"[Title/Abstract] OR "Leishman*"[Title/Abstract] OR leprosy[Title/Abstract] OR "mycobacterium lepr*"[Title/Abstract] OR Hansen[Title/Abstract] OR "lymphatic filariasis"[Title/Abstract] OR "brugia malayi"[Title/Abstract] OR "wuchereria bancrofti"[Title/Abstract] OR onchocerciasis[Title/Abstract] OR onchocerca*[Title/Abstract] OR "river blindness"[Title/Abstract] OR mycetoma[Title/Abstract] OR chromoblastomycosis[Title/Abstract] OR podoconiosis[Title/Abstract] OR scabies[Title/Abstract] OR yaws[Title/Abstract] OR treponem*[Title/Abstract] OR "guinea worm"[Title/Abstract] OR dracunc*[Title/Abstract] OR tungiasis*[Title/Abstract]) AND | AND |
| Methods of interest | (("map"[Title/Abstract] OR "mapping"[Title/Abstract] OR "distribution"[Title/Abstract] OR "niche"[Title/Abstract] OR "suitability"[Title/Abstract] OR "geostatistic"[Title/Abstract] OR "ecolog*"[Title/Abstract]))) AND (model[Title/Abstract])) | AND |
| Publication Limits | (("2000/01/01"[Date - Publication]: "2025/09/30"[Date - Publication])) | |

**Table 2. Inclusion and Exclusion Criteria.**

| Category | Inclusion criteria | Exclusion criteria |
|---|---|---|
| Target Diseases | Skin NTDs in humans: Buruli ulcer, dracunculiasis (Guinea-worm disease), cutaneous leishmaniasis, leprosy, lymphatic filariasis, mycetoma, chromoblastomycosis, onchocerciasis, podoconiosis, scabies, tungiasis, yaws. | Disease not affecting the skin or not an NTD. Studies that predict distributions of skin NTD vectors or non-human reservoir only. |
| Data Source | Use of epidemiological data from surveys, literature, active case searches, or routine surveillance. | Simulated data |
| Modelling Focus | Using spatial or ecological niche models to predict risk, suitability, occurrence, prevalence, or incidence in areas without observed data. | Studies that do not predict the distribution of skin NTDs, for example: statistical/ epidemiological models without spatial component, time series with no spatial element. Studies that analyse associations (e.g., spatial lag, Moran's I) or smooth rates (e.g., Bayesian smoothing) without predicting into unobserved areas. |
| Spatial Scale | Fine- resolution: pixel-level or the lowest administrative unit of a country. | Larger administrative areas such as districts. |
| Publication characteristics | Studies published in peer-reviewed journals or conference proceedings; published between 2000 and 2024 | Not published in peer-reviewed journals or conference proceedings; published before 2000 or after 2024. |

We excluded studies not in peer-reviewed journals, not in English, French, Portuguese, Turkish or Spanish, or published before 2000 or after 2024. We also excluded studies which used spatial analysis without predicting new areas and focusing on vectors and non-human hosts.

unit-level). The diseases of interest were BU, dracunculiasis (Guinea-worm disease), CL, Hansen's disease (leprosy), LF, mycetoma, chromoblastomycosis, onchocerciasis (river blindness), podoconiosis, scabies, tungiasis, and yaws. We included studies that modelled either prevalence or suitability, using statistical models, machine learning algorithms, or geostatistics.

## Study selection

All retrieved articles were uploaded to the Rayyan software, (S1 File). Two reviewers (MAB and RH) independently screened the titles and abstracts of each article to determine its relevance to the research topic. Full-text versions of articles which passed this initial screening were retrieved and independently evaluated by the same two reviewers. Any disagreements were resolved through discussion or, when necessary, by consulting a third reviewer (HS).

## Data extraction process

Data were extracted independently by two reviewers using a bespoke Microsoft Excel spreadsheet containing the following fields: disease of interest, study region, country of lead author's institutional affiliation, source of epidemiological data, model type(s), software used, environmental covariates, process to select predictors, sensitivity analysis, performance evaluation metrics, availability of model outputs, availability of model code, challenges and limitations of modelling approaches, and potential applications of model outputs. The source of epidemiological data was categorised as case notifications (defined as formal reports of disease detection from health facilities or laboratories to a central surveillance system), passive surveillance (defined as records of cases collected from health facilities), case search (defined as detection from a dedicated case finding activity for which the denominator population was not defined), surveys, and literature review (S2 File).

## Risk of bias assessment

We conducted a structured risk of bias (RoB) assessment for all 68 included studies using a bespoke assessment tool incorporating elements from the Joanna Briggs Institute (JBI) checklist for cross sectional studies and a checklist for maximising reproducibility of ecological niche models [32]. Each study was evaluated across ten methodological domains, covering clarity of inclusion criteria, definition of study area and population, representativeness of participants, case definition, validity of exposure measurement, handling of environmental covariates, treatment of background or pseudoabsence data, and approaches to sampling bias, spatial autocorrelation, and statistical analysis. Based on these assessments, studies were categorized as low, moderate, or high risk of bias (S3 File).

## Synthesis methods

Data were synthesised narratively, and key results were tabulated and visualised using plots produced in R (R version 4.1.1 (2021-08-10) [33].

## Results

The literature search identified 1,756 studies relevant to the search terms. After duplicate removal, 818 articles were screened by title and abstract, narrowing the selection to 126 studies eligible for full-text review (Fig 1). Fifty-seven studies were excluded because they did not make spatial predictions (n = 39), did not use human cases (n = 15), did not make fine scale predictions (n = 1) or were not primary research studies (n = 1)). Ultimately, 68 studies met the inclusion criteria and were included in the final analysis.

The following sections present the findings, focusing on the key themes that emerged from the reviewed literature. Key results are shown in Table 3.

## Diseases of Interest

The studies examined a range of skin NTDs, with different transmission modes and types of disease agents (Fig 2). Vector-borne diseases, including CL, LF, and onchocerciasis, accounted for most studies (51 out of 68. The highest number of studies focused on CL (26 studies; records 1–26 in S2 File), followed by LF (18 studies; records 27–44 in

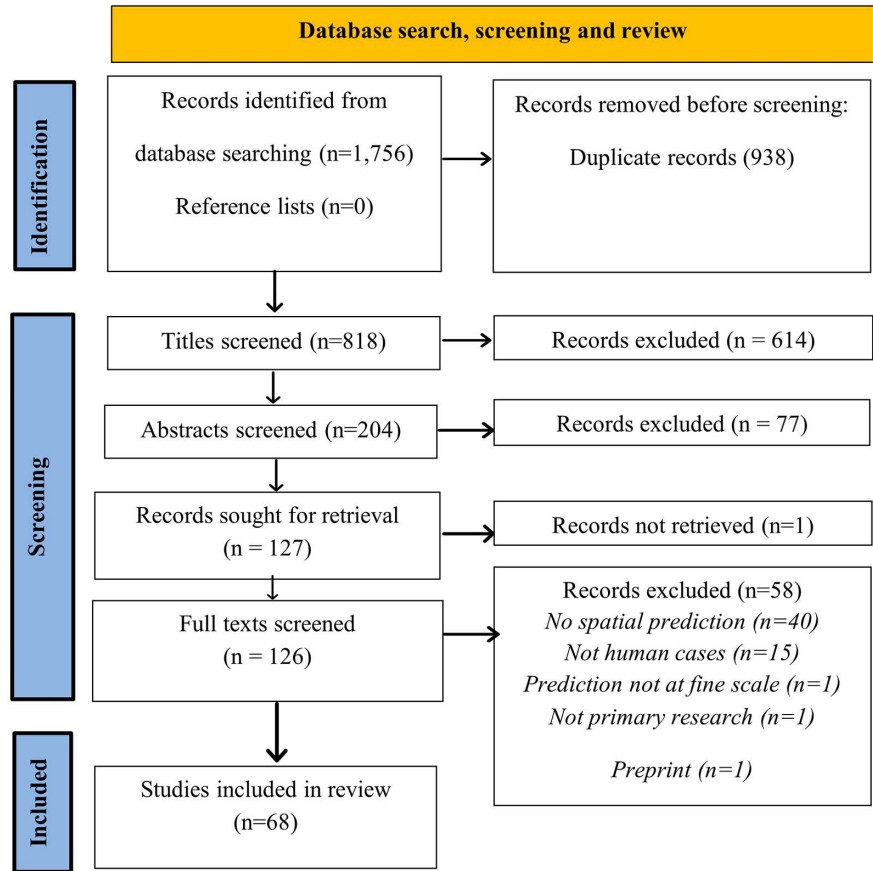

**Fig 1. Flow diagram illustrating the study selection process.**

(S2 File) and onchocerciasis (7 studies). Podoconiosis was the most commonly modelled among non-vector-borne conditions examined (7 studies; records 52–59 in Supplementary (S2 File), while mycetoma (bacterial or fungal origin) [34–36] and tungiasis were each studied in three cases [37–39]. Among bacterial diseases, BU appeared in two studies [40, 41], while lepros [42] was modelled in one [43]. One study modelled Dracunculiasis, caused by a parasite [44]. Notably, no studies were found for scabies, yaws, or chromoblastomycosis, presumably reflecting uncertainty about the type and strength of environmental drivers, and the multitude of causative organisms, in the case of chromoblastomycosis.

## Study regions

The studies covered many regions, focusing on areas where skin NTDs are endemic, including Africa, Asia, North and South America, and parts of Europe (Fig 3). Although Turkey occupies an intermediate, transcontinental position and has a high Human Development Index, it was considered Global South for the purposes of this study. Africa was most prominently represented, with several countries showing high frequencies of studies. The highest number of studies (n = 19) were from Ethiopia, followed by Nigeria (n = 16), Kenya (n = 14), Sudan (n = 14), and Côte d'Ivoire (n = 14), as indicated by the darkest shades on the map in Fig 2. Outside Africa, Brazil (n = 11) and Sri Lanka (n = 8) were the only countries with more than six studies.

While the vast majority of study countries were in the Global South, more than half of the studies (36 of 68) were led by authors from Global North institutions (Fig 4). Nevertheless, several studies were led by authors from Global South institutions in endemic regions such as Ethiopia (7 studies), Turkey, Sudan, Colombia, and Brazil.

**PLOS Neglected Tropical Diseases**

**Table 3. Summary of Results.**

| Data Extraction Category | Frequencies |
|---|---|
| Total Studies Included | 68 |
| Diseases of Interest | CL (26), LF (18), Onchocerciasis (7), Podoconiosis (7), Mycetoma (3),Tungiasis (3), BU (2), Leprosy (1), Dracunculiasis (1) |
| Study Countries | Ethiopia (19), Nigeria (16), Kenya (14), Sudan (14), Côte d'Ivoire (14), Brazil (12), Sri Lanka (8) |
| Country of institutional affiliation of lead authors | UK (16), USA (10), Ethiopia (7), Turkey (4), Australia (3), Colombia (2), Sudan (2), Switzerland (2), Brazil (2), Pakistan (2), Iran (2), Sweden (1), Sri Lanka (1), Hungary (1), Ghana (1), Zambia (1), Egypt (1), Mexico (1), France (1), Nigeria (1), Tunisia (1), Denmark (1), Indonesia (1), Palestine (1), Qatar (1), Argentina (1), Japan (1) |
| Modelling Approaches | Geostatistical (35: GLMMs, IDW, Kriging). Ecological (33: GLMs, Maxent, RF, etc.) |
| Source of Epidemiological Data used for modelling | Survey (33), Passive (22), Notification (21), Literature (17), Case Search (3) |
| Inclusion of True absence (TA) or Pseudoabsence (PA) Data | PA (35), PA & TA (5), TA only (20), None/NR (11) |
| Software Used | R (39), GIS tools (32), MaxEnt, STATA/SPSS (4), WinBUGS/GeoBUGS (4) |
| Environmental Datasets Used for Modelling | Precipitation (60), Temp (56), Elevation (60), Pop. Density (30), LULC (27), NDVI/LST (28), Water (28), Humidity (28), Soil (25) |
| Socioeconomic and development-related covariates | Night-time light (20), Livestock (5), Healthcare access (2), Program indicators (2), Conflict indicator (1), food security indicator (1) |
| The Process of Selecting Predictors in the Study | Literature(14), Correlation(16), Variable importance(12), Stepwise (AIC) (15), Expert judgment(11) |
| Sensitivity Analysis | Parameter variation, Uncertainty quantification, Threshold sensitivity |
| Methods to Evaluate the Performance of Models | AUC/ROC, Cross-validation, Fit stats, Residuals, External validation (17 studies) |
| Challenges & Limitations to effective modelling | Data gaps, Generalizability, High computing cost, Code sharing, Sampling bias |
| Model Applications in Public Health | Hotspot identification, MDA planning, Integration, Targeting Surveillance and interventions |

## Modelling approaches used

Of the studies included in the review, 35 were classified as geostatistical models, as they explicitly modelled spatial dependency in some way. These models commonly incorporated spatial correlation structures and environmental factors. Among them, 15 studies used generalized linear mixed models (GLMMs), often within Bayesian or maximum likelihood frameworks (S2 File). A smaller number of geostatistical studies employed interpolation techniques such as inverse distance weighting (IDW; used in five studies), kriging (four), and spline interpolation (three), typically without adjusting for environmental covariates. Ecological models, defined here as those predicting predict disease suitability based on environmental covariates, were used in 33 studies. Among these, generalized linear models (GLMs) were the most widely used method, appearing in 23 studies, followed by maximum entropy (MaxEnt) in 16, and generalized boosted regression trees (BRTs) in nine. Random forest (RF) methods were applied in 11 studies, while Generalized Additive Models (GAMs) were used in ten. Less commonly used ecological approaches included Multivariate Adaptive Regression Splines (MARS) in eight studies and Artificial Neural Networks (ANNs) in six.

## Source of epidemiological data

Epidemiological data used in the studies were derived from various resources, reflecting the diverse approaches to gathering health information (Fig 5). The most common source of data was survey data, used in 33 studies (21

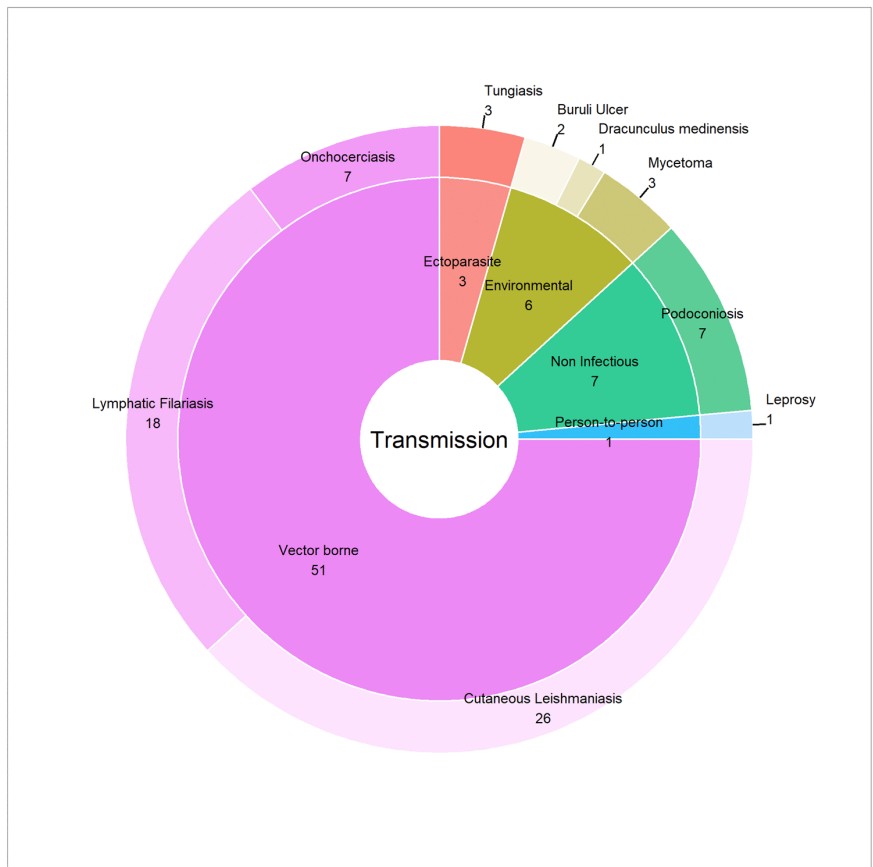

**Fig 2. Number of Studies Identified by Disease and Transmission (Total = 68).**

spatial and 12 non-spatial). Data originating from passive surveillance (health facility records) were used in 22 studies (14 ecological and eight geostatistical), and case notifications originating from national reporting systems were used in 13 ecological and eight geostatistical studies. Twelve ecological and five geostatistical studies used data captured from literature reviews. The least utilized source was case search, which was reported in only three non-spatial studies. Data sources were multiple for some studies. One study did not report the type of data used for modelling [45].

### Inclusion of background or pseudoabsence data

Researchers used background or pseudoabsence points to improve model accuracy and compensate for the lack of reliable true absence data. Of the 68 studies reviewed, 35 used pseudoabsence in place of true absences, and four used both pseudoabsence and true absences, making a total of 40 studies that incorporated pseudoabsence (S2 File). This practice was common in ecological models and some hybrid models that integrated ecological and statistical methods. The most frequent method for generating background data was the random selection of points across the study area, noted in 10 studies. Typically, between 5,000 and 10,000 background points were randomly distributed, sometimes from within buffer zones around presence locations to minimise sampling bias. In five studies, the use of MaxEnt software suggested a default reliance on random background sampling, even when not explicitly

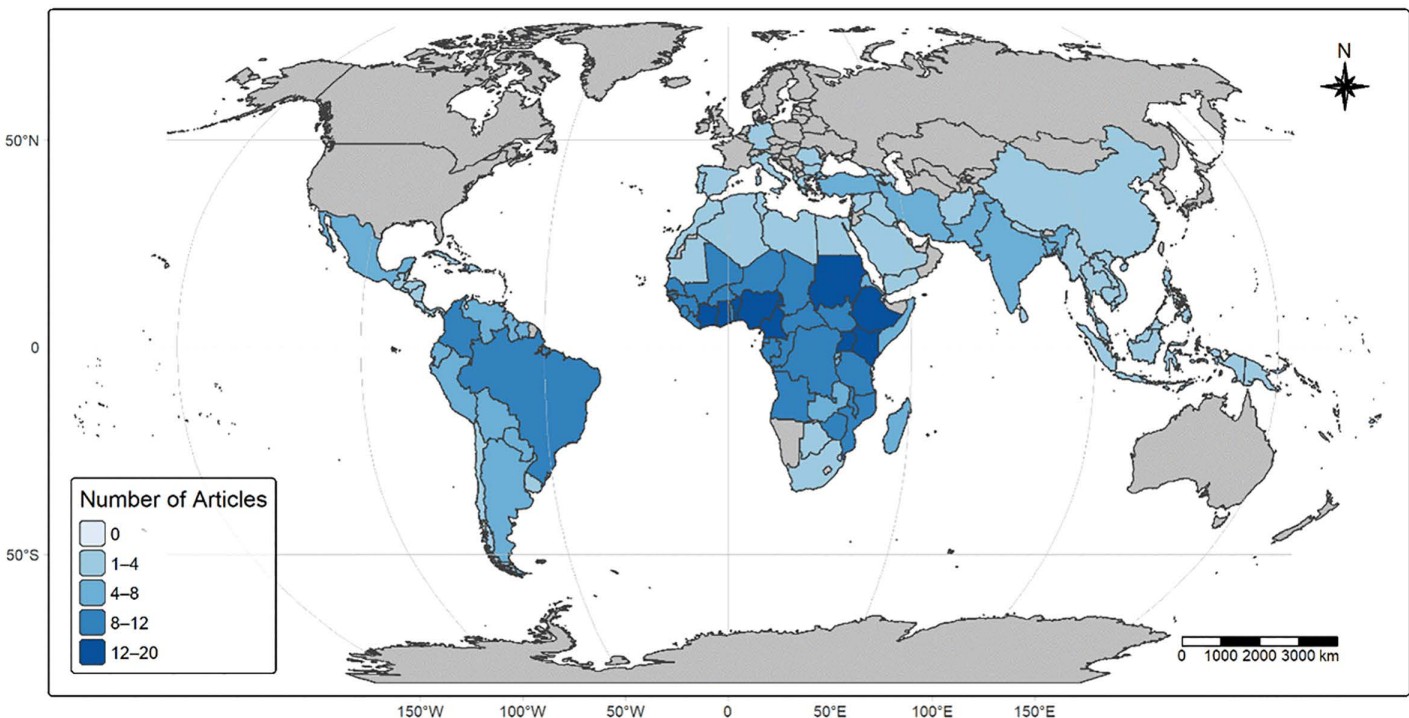

**Fig 3. Number of skin NTD modelling studies identified by country.** The colour gradient on the map, ranging from (light to dark blue) visually conveys the number of studies per country. **Basemap source:** Natural Earth, 2024 (http://www.naturalearthdata.com). License: Natural Earth data are in the public domain and freely available for any purpose.

stated. In six studies, pseudoabsence data were generated using more systematic or informed approaches. Three studies applied probabilistic methods informed by evidence consensus layers, allowing for targeted selection of pseudoabsence points in regions with lower evidence of disease presence. Two studies used the Surface Range Envelope (SRE) approach to exclude environmentally suitable areas from pseudoabsence generation. One study modelling LF employed expert knowledge to avoid selection of pseudoabsence points from endemic regions or those environmentally suitable for vectors, using land cover classifications and aridity indices. Furthermore, three studies incorporated sampling bias correction techniques, including selection of background points in proximity to occurrence data using a kernel density approach, or using a bias layer representing sampling intensity. Lastly, one geostatistical study introduced zero-prevalence data points in non-endemic areas to account for the high number of zero values and reduce overprediction along the edges of disease-free zones.

## Software applied

A wide range of software tools was employed, reflecting diverse and advanced analytical approaches. Forty studies utilised R Statistical Software, leveraging specialized packages for modelling and visualization. Twenty-nine studies used Geographic Information System (GIS) tools, such as ArcGIS and QGIS, GRASS, and Google Earth, which were commonly used for spatial analysis and visualisation, often in combination with R Statistical and MaxEnt-supported species distribution modelling. STATA and SPSS were utilised in three studies, either as a standalone tool or in combination with R. WinBUGS or GeoBUGS were applied in four studies to conduct Bayesian spatial analyses.

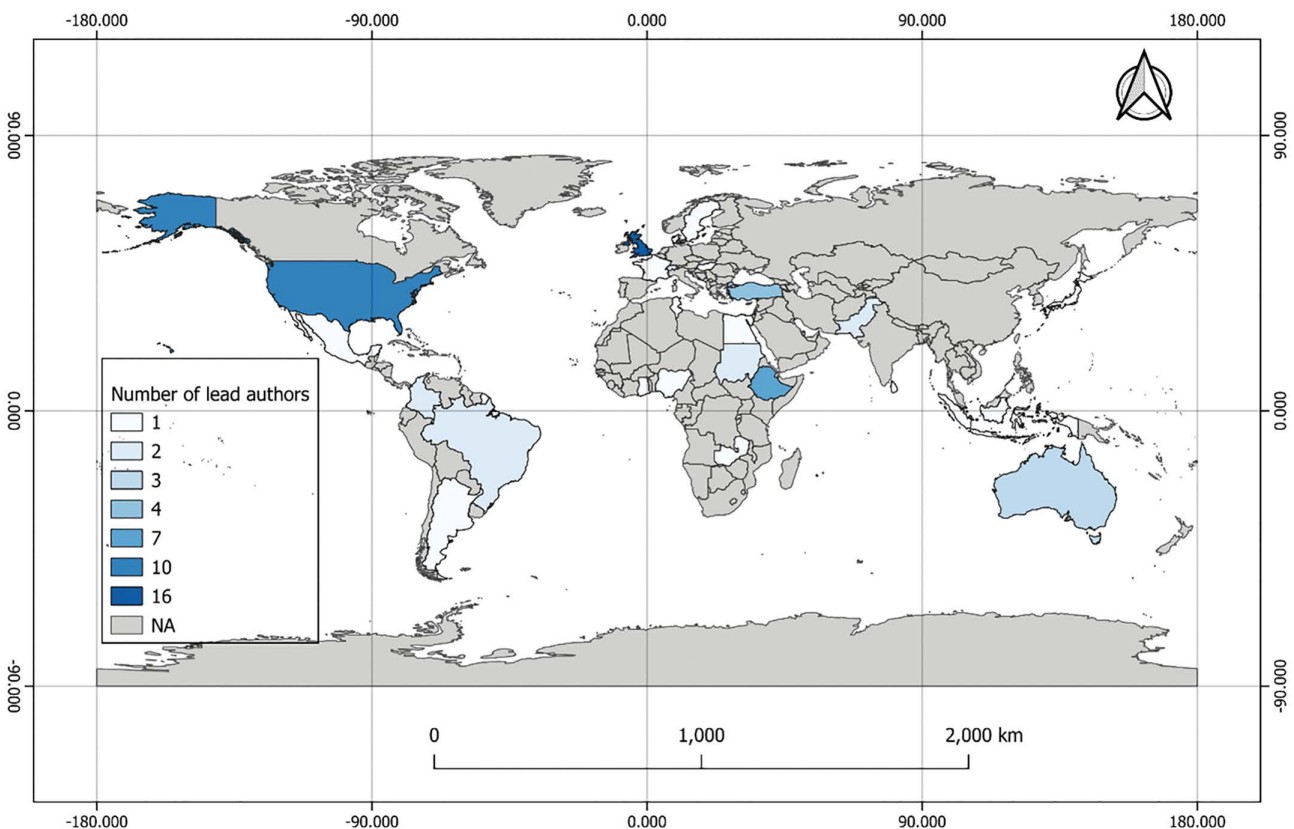

**Fig 4. Institutional affiliations of lead authors, by country.** Basemap source: Natural Earth, 2024 (http://www.naturalearthdata.com). License: Natural Earth data are in the public domain and freely available for any purpose.

## Environmental datasets used for modelling

The studies included a variety of environmental datasets that were integrated into models to predict disease distributions (Fig 6). Precipitation was the most commonly used predictor, included in 60 studies. This was followed closely by temperature in 56 studies, highlighting the central role of climatic variables in disease distribution. Topographical factors, including elevation, slope, aspect, and terrain roughness, were used in 50 studies. These were notably applied in studies on CL (n = 17), LF (n = 15), and podoconiosis (n = 7), providing context for understanding the habitats of disease vectors and the spatial distribution of NTDs. For instance, studies in Ethiopia used elevation datasets to estimate podoconiosis cases and assess how altitude influences disease prevalence [46]. Human population density was used in 30 studies, and vegetation indices such as NDVI and Land Surface Temperature (LST) were used in 29 studies. Land use and land cover (LULC) data were used in 28 studies, while datasets representing proximity to water bodies or coastlines were also included in 28 studies. Atmospheric conditions such as aridity and humidity were used in 25 studies, and soil properties were considered in 26 studies, particularly for modelling podoconiosis, where soil composition, structure, and physical characteristics are directly linked to disease risk [47,48].

Less commonly used predictors included development indicators, such as night-time light emissivity (16 studies), proxies of exposure (seven studies), livestock density (three studies), and healthcare access and programmatic indicators, each reported in only two studies. The diversity in selected variables reflects a multidimensional approach to understanding disease dynamics across varied geographic, ecological, and socioeconomic contexts. Many of the climatic and

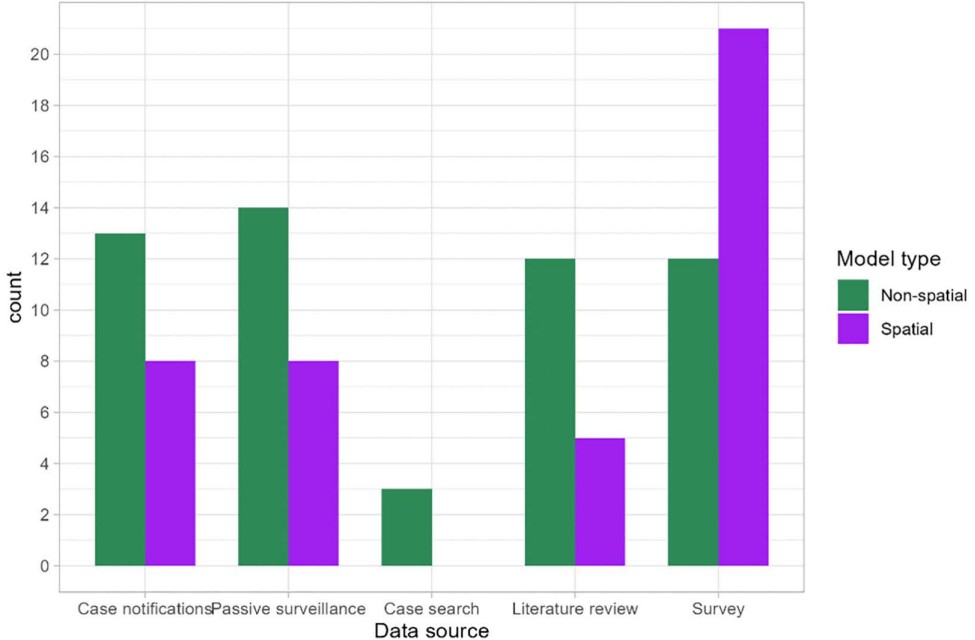

**Fig 5. Source of epidemiological data used for modelling the distribution of skin NTDs.**

environmental datasets, such as rainfall, temperature, humidity, solar radiation, NDVI, LST, and elevation, were obtained from global sources like WorldClim and satellite-based climate reanalysis products.

## The process of selecting predictors

In most studies, the process of selecting predictors began with a literature review, drawing on existing research to identify environmental and socio-demographic factors associated with disease transmission and distribution (S2 File). This often served as the foundation for assembling an initial pool of relevant variables. Following this, correlation analysis was commonly used to assess relationships among the selected variables. Highly correlated predictors, such as elevation and temperature variables, are carefully considered, with one often removed to reduce collinearity. In certain cases, variable importance measures were applied, especially when using machine learning models like Random Forests. These models rank variables based on their influence within a model, further refining the selection process by eliminating those with limited impact. Some studies used stepwise selection techniques, including those based on the Akaike Information Criterion (AIC). This iterative approach involves adding and/or removing variables based on their statistical influence, ensuring that only the most significant predictors are retained. Finally, expert judgment complemented the statistical techniques throughout the process. Informed by contextual and field-based knowledge, domain experts validated or retained variables that may not have appeared significant in quantitative analyses but held epidemiological relevance, ensuring the final model reflected the most meaningful drivers of disease distribution.

## Sensitivity analysis

Our findings indicated that several sensitivity methods are typically used to assess how robust the model is under varying conditions. One common approach identified was parameter variation, especially in models including MaxEnt and Random Forest. By adjusting key parameters, researchers can test model sensitivity to various inputs and identify which ones are driving the results and which have less influence. Threshold sensitivity in binary classification models was frequently

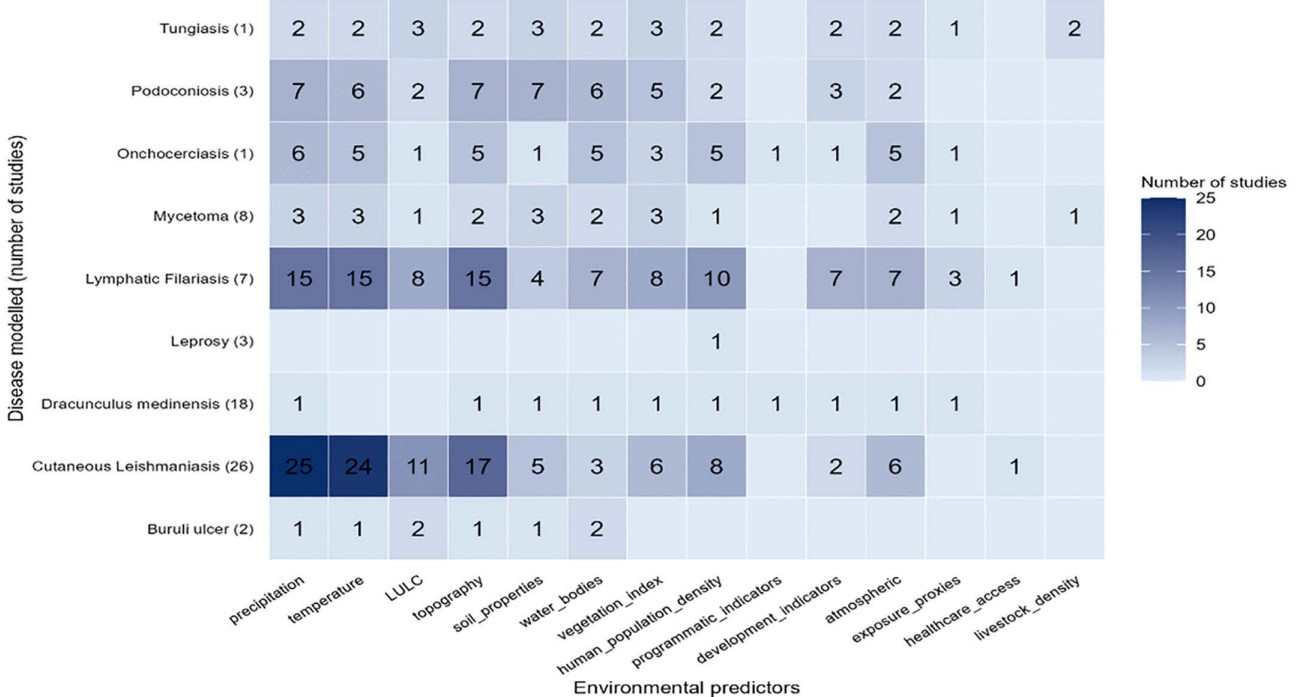

**Fig 6. Environmental Datasets Used for Modelling.**

highlighted as a critical consideration. Researchers adjusted decision thresholds and analysis to evaluate how changes influenced the accuracy of predictions. This ensures transparency around the impact of the threshold on predictions.

## Methods to evaluate the performance of models

A variety of methods were used to evaluate model performance (S2 File). These include uncertainty quantification (widely used to evaluate geostatistical models), the Receiver Operating Characteristic (ROC) Curve and Area Under the Curve (AUC), cross-validation, goodness-of-fit statistics, classification performance metrics, posterior predictive checking and model convergence, autocorrelation and spatial plots, jackknife tests and variable importance, probability-based evaluation, residual diagnostics, and variograms. A total of seventeen papers mentioned some form of external validation, often referred to as cross-validation, test datasets, out-of-sample predictions, or holding out data for validation. Many studies employed k-fold cross-validation or spatial block cross-validation, where the data are split into subsets to test the model on unseen data, ensuring spatial or temporal independence between training and testing datasets. Additionally, six studies compared model predictions with real-world survey data from locations not included in the model training.

## Challenges and limitations of modelling approaches

The review identified several challenges in modelling approaches for predicting skin NTDs. A key issue was the limited availability and inconsistency of epidemiological data, often due to weak health surveillance systems. Additionally, the generalizability of models poses a challenge, as those developed for one region may not apply to others due to varying environmental, social, and demographic factors. Some techniques, such as Bayesian Geostatistical Modelling, require significant computational resources, which can be a barrier in resource-limited settings. Furthermore, the ecological complexity of NTDs, shaped by factors like human behaviour, vector biology, and climate, adds to the modelling difficulties.

These challenges underscore the need for improved data collection, the inclusion of diverse covariates, incorporation of techniques to account for variation in surveillance intensity, and robust validation processes to enhance predictive accuracy and relevance across different contexts.

### Risk of bias assessment

Across the 68 studies assessed, 23 (35%) were rated low risk of bias, 26 (38%) moderate, and 19 (27%) high (S3 File). While most studies clearly defined study areas and applied appropriate analyses, weaknesses were common in participant representativeness, sampling bias, and the handling of spatial autocorrelation within ENMs. Geostatistical approaches tended to achieve lower risk ratings (53% rated as low and 15% rated as high), whereas ecological niche modelling studies were more often classified as high risk (12% rated as low and 42% rated as high).

### Discussion

This systematic literature review aimed to examine and synthesise modelling approaches used to predict the distribution of skin NTDs, with a focus on identifying predominant methods, evaluating their applications, and highlighting research gaps. A total of 68 studies were included.

Studies were geographically diverse, with a large proportion of studies conducted in Africa. Lead author institutions were almost evenly distributed between the Global North (36 studies) and the Global South (32 studies). Although the UK and USA had the highest number of studies by lead author, endemic countries such as Ethiopia, Turkey, Sudan, and Brazil are leading their own modelling efforts.

Most studies focused on vector-borne diseases (VBDs), such as CL, LF, and onchocerciasis. The review revealed significant diversity in model types, with both geostatistical and ecological models used extensively. The studies used different types of epidemiological data, most commonly survey and passive surveillance data, and the selection of covariates varied depending on the disease, with climate, elevation, land cover, and soil type being commonly applied predictors.

The key finding of this review is the dominant use of geostatistical models in the prediction of skin NTD distributions, especially GLMMs for capturing spatial patterns. These models, including Bayesian and generalised linear mixed models, explicitly incorporate spatial dependency and provide more robust predictions for diseases with continuous prevalence outcomes. This is consistent with approaches used in other NTDs, such as schistosomiasis and STHs, and reflects an increasing recognition of the importance of spatial autocorrelation in disease mapping [49]. These studies most often used high-quality survey data as a primary data source, indicating a preference for higher-quality, structured data in spatial modelling efforts.

In addition, ecological models continued to be used, especially in studies relying on presence-only data from health facility reports, case notifications, or extracted from the literature. These models do not account for spatial structure but are widely used due to potential application without true absence data. Ecological models addressed data limitations by incorporating pseudoabsence points, often using random or evidence-informed sampling. While flexible, this can limit reliability, reduce accuracy, and even exacerbate biases. The majority of ecological models in this review used spatially random pseudoabsence points, which can mis represent areas which are suitable for a disease but in which it has not yet emerged, or has not been detected. Depending on the context and study area, alternative approaches may be more appropriate. The use of evidence consensus layers to select pseudoabsences in areas less likely to be endemic can reduce the risk of pseudoabsences being selected from within suitable areas [40]. The surface range envelope approach, which restricts pseudoabsences to areas that are environmentally distinct from occurrence locations, can also help to reduce this effect. However, if sampling was targeted (intentionally or unintentionally) to locations with particular environmental characteristics, this can exacerbate the impact of selection bias [50]. Studies included in this review used environmental and climatic variables, especially precipitation and temperature, which were the most used predictors across the studies included. These environmental factors are known to significantly influence the distribution of vector-borne and

environmentally mediated diseases. Other commonly used predictors included elevation, land cover, vegetation indices, and soil characteristics, particularly in podoconiosis models [51]. The selection process for predictors was typically informed by prior literature, expert judgement, and correlation testing.

Although in risk of bias assessment about one-third of the studies were classified as low risk, the majority were rated as moderate or high risk, reflecting recurring weaknesses in sampling, spatial autocorrelation, and reliance on presence-only data, these limitations undermine confidence in outputs and their policy relevance. In contrast, studies applying rigorous geostatistical methods showed stronger quality, underscoring the importance of standardized methodologies and transparent reporting practices.

The models reviewed have important practical applications. These models can help target interventions by identifying high-risk areas, allowing health authorities to direct resources efficiently. For example, in Ghana, spatial models predicted schistosomiasis prevalence [52], allowing for targeted MDA campaigns. This was achieved through Model-Based Geostatistics (MBG), integrating environmental factors and parasitological data, demonstrating how models can support effective resource allocation for NTD control [53]. To support the uptake of models for public health planning, authors should engage with decision makers to identify the most useful presentation of estimates. For example, summaries such as population living in suitable areas or number of people infected are likely to be more useful for resource allocation than continuous surfaces at national or continental level. For increased interpretability of continuous outputs, authors could present these at lower levels in supplementary material, or even better, as an interactive application allowing users to zoom in on locations of interest [54]. Prediction outputs for different skin NTDs could also be overlaid to show overlapping risk areas, identifying targets for integrated interventions, as promoted by the WHO skin NTD strategy[2]. However, this may be limited by the diversity of approaches used for modelling different diseases. End-users' confidence in model outputs may be limited by their understanding of the methodologies, trust in the data used, and prediction accuracy. Increased transparency in the reporting of methods and results would go some way towards addressing this. We identified studies which did not report key components including the source of epidemiological data, the approach to pseudoabsence generation, the spatial resolution, and prediction uncertainty, and only seven of 68 publications made their model code available. Adoption of minimum reporting standards checklists for ecological niche models and geostatistical models would enhance transparency and trust in outputs. Despite promising developments, there are multiple challenges in developing and applying models for skin NTDs. A key challenge lies in the limited and often inconsistent epidemiological data across regions, largely resulting from underreporting and inadequate surveillance systems. The lack of true absence data and the reliance on pseudoabsence points introduce additional uncertainty into models [55]. Furthermore, the complex interplay of environmental, biological, and socio-economic drivers of skin NTDs makes them challenging to model accurately. Models developed using data from one region may lack generalizability to others due to environmental and socio-demographic differences. Key strengths of this review include a comprehensive search strategy and a double-review process for screening and data extraction, which adds rigor and reduces bias. However, it also has limitations, such as the exclusion of grey literature and potential publication bias. The heterogeneity of included studies, in terms of model types, data sources, and outcomes, makes it difficult to perform direct comparisons or quantitative synthesis.

## Conclusions

Modelling of skin NTDs is an emerging and growing field with substantial potential for supporting control and elimination strategies. However, the wide variation in methodologies and input data quality suggests a lack of cohesion and best practice across studies. Although these models can yield highly informative outputs, their utility hinges on the credibility and transparency of the modelling methodologies employed. For future research, there is a need to develop models that incorporate more comprehensive environmental, biological, and socio-economic covariates, as these are critical to understanding the complex transmission dynamics of skin NTDs. Increasing access to model codes and encouraging the use of more advanced modelling techniques, such as Bayesian Geostatistical Models, could also lead to improvements

in predictive accuracy. Collaborative efforts between researchers, governments, and public health organizations will be essential to overcome data limitations and extend the reach of these modelling tools into endemic regions. With improved transparency and collaboration, predictive models can meaningfully contribute to WHO's 2030 NTD elimination targets. Strengthening modelling capacity in endemic regions through equitable collaborations and sustained investment will be critical to achieving scientific equity and ensuring that predictive models directly inform national disease control programs.

## Registration and protocol

The protocol for this systematic review was registered with the **Open Science Framework (OSF)** (registration https://doi.org/10.17605/OSF.IO/H6B5N). The full protocol is publicly accessible at the registration link.

## Supporting information

**S1 File. Master list of all studies identified (n = 818) after duplicate removal.**
(XLSX)

**S2 File. Supplementary Table.** Extracted data for the 68 included studies.
(DOCX)

**S3 File. Supplementary Table.** Risk of bias and quality assessments for the 68 included studies.
(DOCX)

**S4 File: Supplementary PRISMA 2020 Checklist (PRISMA 2020 checklist SRV RA025.docx).** Completed PRISMA 2020 checklist indicating where each reporting item is addressed in the manuscript.
(DOCX)

## Author contributions

**Conceptualization:** Mesoud A.Bushara, Hope Simpson.

**Data curation:** Mesoud A.Bushara, Rowa Hassan, Hope Simpson.

**Formal analysis:** Mesoud A.Bushara, Hope Simpson.

**Methodology:** Mesoud A.Bushara.

**Supervision:** Rana Ahmed, Eltayeb Ganawa, Hope Simpson.

**Visualization:** Mesoud A.Bushara.

**Writing – original draft:** Mesoud A.Bushara.

**Writing – review & editing:** Mesoud A.Bushara, Rowa Hassan, Rana Ahmed, Jorge Cano, Gail Davey, Eltayeb Ganawa, Hope Simpson.

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
